# FUNCTIONAL LEARNING-MODULES FOR ATARI BALL GAMES

**Andrew Melnik, Sascha Fleer, Malte Schilling & Helge Ritter**
CITEC, Bielefeld University, Bielefeld, Germany
{anmelnik, sfleer, mschilli, helge}@techfak.uni-bielefeld.de

## ABSTRACT

We present a cognitive architecture capable of end-to-end learning across different but conceptually similar Atari games using the same learning algorithms, network architecture, and hyperparameters. The cognitive architecture uses object-based representations to generalize across different games and consists of a pipeline of functional modules. Each module is allocated for a specific functionality, but exact policies and input-output relationships are formed by reinforcement learning and supervised learning algorithms. The convergence rate of the cognitive architecture is considerably faster than deep reinforcement learning architectures that rely on pixel-based information and follow a purely end-to-end approach. Moreover, the modules of the cognitive architecture can be directly reused in different game environments without retraining. Our approach is inspired by the modular structure of biological brains, where functional modules evolved, but still can be shaped through learning from new experiences.

## 1 INTRODUCTION

Essential ingredients of learning are the priors that we put into the machine and the techniques we use for optimization and search (Bengio, 2013). Evolutionary development shapes priors in biological living systems. These priors allow biological brains to exploit rewarding nuances of an environment upon first discovery (Blundell et al., 2016). Nevertheless, these priors leave a space for behavioral choice and adaptation. Biological agents do not learn by optimization of a uniform and undifferentiated neural network (Hassabis et al., 2017). Instead, they are modular with interacting subsystems working in parallel.

Reinforcement learning (RL) techniques combined with deep neural nets (DNN) have shown to be effective in action optimization in complex tasks (Kidzinski et al., 2018). As one example, deep Q-network (DQN) with pixel based input is capable of end-to-end learning of optimal behavior in complex novel environments. However, modern deep reinforcement learning (DRL) algorithms require millions of interactive steps and repetitions to attain human-level performance in straightforward game environments (Blundell et al., 2016).

Recent AI research attempts to overcome these learning limitations of DRL networks by developing cognitive architectures that mirror the complementary learning systems in mammalian brains (Chang et al., 2016; Hamrick et al., 2017), e.g., by using a factorization of a physical scene into object-based representations and a neural network architecture capable of learning pairwise interactions. This allows for quick adaptation to specific object properties, to novel changes in an environment, to dynamics of different environments, and it allows for knowledge transfer from one environment to another (Chang et al., 2016; Kansky et al., 2017).

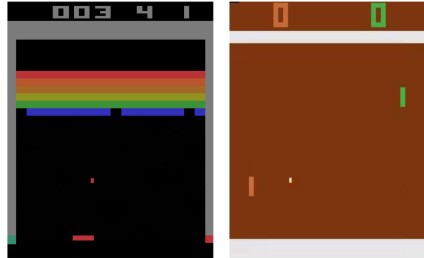

Figure 1: We tested the cognitive architecture in the OpenAI-Gym Breakout and Pong environments

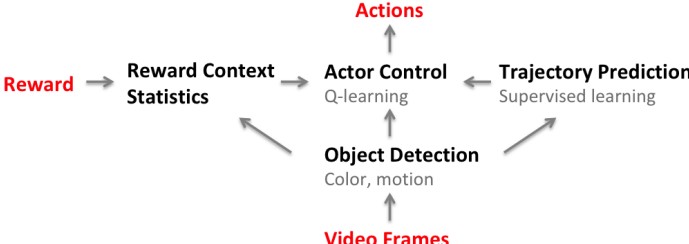

Figure 2: The block scheme of the proposed cognitive architecture with four functional modules.

## 2 FUNCTIONAL MODULES

Our approach reflects the modular structure of biological brains, which is assumed as pre-given and shaped by evolution. But the individual modules are capable of learning through new experiences. We tested the cognitive architecture in the OpenAI-Gym-Atari Breakout[1] and Pong[2] environments (Fig. 1) (Brockman et al., 2016). The cognitive architecture uses object-based representations and consists of a pipeline of functional modules (Fig. 2). Each module is allocated for a specific functionality, but exact policies and input-output relationships are formed by reinforcement learning and supervised learning algorithms. The ball and the paddle have similar dynamics in the Pong and Breakout environments. Therefore, the cognitive architecture is capable of reusing learned dependencies over these game environments. In the current architecture, the following modules are applied:

**Object Detection.** For object detection, we use color and movement information (difference between sequential frames). A state vector comprises properties of an object at a given time instance (position, velocity, object type, object size, and applied action). Further processing of the state vectors occur in successive functional modules.

**Trajectory Prediction.** The input consists of a sequence of recently observed spatial two-dimensional Euclidean coordinates of the ball. We use one-hot encoded vectors for the representation of these coordinates. The module learned to predict the intersection point of the ball with the paddle line via supervised learning. In both games, the paddle line is the region where the paddle and the ball are able to collide.

**Actor Control.** A state vector of the paddle comprises previous actions applied to the paddle, in addition to the distance between the current position of the paddle and the target-position specified by the Trajectory-Prediction module. We use tabular Q-learning for training. Instead of using the game-score as a reward indicator, a context based, state-dependent reward value is provided by the Reward-Context-Statistics module.

**Reward Context Statistics.** The module collects statistics on collision events between the ball and the paddle objects and related reward events. Based on these data it provides a regression between the distance value in the state vector and expected reward value.

---

[1]https://en.wikipedia.org/wiki/Breakout_(video_game)
[2]https://en.wikipedia.org/wiki/Pong

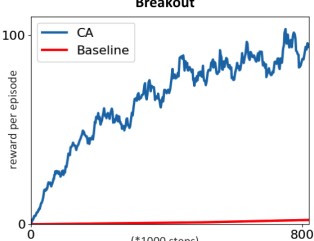 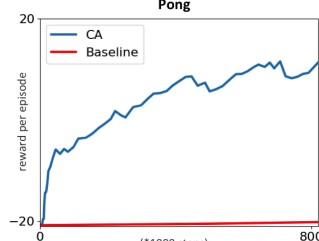

Figure 3: We compared performance of the cognitive architecture (CA) against other baselines. The Breakout baseline is taken from Fig. 4: LSTM, 0.95 in (Vezhnevets et al., 2017), and it requires about 3.5*e7 steps to achieve the score of 100 points. The CA requires 7.5*e5 steps to achieve the score of 100. The Pong baseline is yielded by running the script provided here (Karpathy, 2016), and it requires about 2*e8 steps to achieve the score of 13. The CA requires 7*e5 steps to achieve the score of 13.

## 3 EXPERIMENTS

**Baselines.** The results demonstrated an advantage in the convergence rate of our cognitive architecture with respect to other baselines as explained in Fig. 3.

**Transfer Learning.** We examined how well the cognitive architecture can transfer existing knowledge to a new game environment. We trained the cognitive architecture on one of the game environments and checked the performance in other game environments. Afterwards, the cognitive architecture was immediately able to perform well in the novel environment without requiring initial training. Thus, another advantage of functional modules is reusability in different environments without initial training. However, this presupposes similar objects' dynamic between the different game environments.

## 4 DISCUSSION

The convergence rate of the presented cognitive architecture with functional modules is considerably faster than with a DQN pixel based approach. We assume that one factor contributing to the boosted learning rate is the disentangled representation of objects' features in the Object-Detection module (Fig. 2). Another contributing factor is the hierarchical nature of the cognitive architecture.

Disentangling what a cognitive architecture can control in an interactive environment and finding latent factors of variation in rewards are difficult tasks (Bengio et al., 2017). In the current implementation of the cognitive architecture, we specified priors for parsing a scene into an object-based representation, as well as the set of provided functional modules. This leads to a much faster convergence rate in comparison to other approaches, but shows limited generalization scopes compared to pixel based end-to-end training. However, it appears likely that only a small set of useful functional modules is required for successful inference in a variety of Atari environments. Therefore, the presented approach with functional modules as priors inbuilt in the cognitive architecture has a potential for further development into a more general cognitive architecture that can be transferred to even more game scenarios.

A possible direction of further development of the cognitive architecture is the deictic approach (Ballard et al., 1997). The approach proposes mechanisms of implicit referencing by pointing movements to bind objects in the world to cognitive programs. Deictic computation provides a mechanism for representing the essential features that link external sensory data with internal cognitive programs and motor actions. This could be integrated as a further module that carries out mechanisms of attention.

### ACKNOWLEDGMENTS

This work was supported by the Cluster of Excellence Cognitive Interaction Technology CITEC (EXC 277) at Bielefeld University, which is funded by the German Research Foundation (DFG).

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
