# OpenReview forum: "Functional Learning-Modules for Atari Ball Games"
_ICLR.cc/2018/Workshop — Reject_

### Official Review · AnonReviewer2 · 2018-03-06

**Rating:** 4
**Confidence:** 4

**Review:**

# Summary
This paper presents an object-interaction-based approach for deep reinforcement learning. The idea is to provide high-level information about object interaction (e.g., velocity/position of balls, the interaction between objects). Experimental results show that the proposed approach outperforms existing image-based deep RL architectures (i.e., CNN).

[Pros]
- A reasonable claim about the necessity of object-based representations for RL.

[Cons]
- Novelty is low.
- The experimental result is not convincing enough.

# Novelty
- Though the main claim about the usefulness of object-based representation for RL is reasonable, the proposed idea is not novel in that they provided high-level features instead of proposing a way of "learning" such object-based representations without domain-specific knowledge.

# Quality
- The experimental result is not much convincing. The proposed method provides high-level features (e.g., the velocity of balls, etc) to the agent, so it is expected to perform better than CNN-based architecture which has to learn from raw pixels. Instead, it would be much more convincing to propose an architecture that captures such object-interaction without any domain-specific knowledge. It would be also interesting to compare against existing methods in terms of generalization performance.

---

### Official Review · AnonReviewer1 · 2018-03-08
**Interesting direction, but confusing exposition of ideas**

**Rating:** 4
**Confidence:** 4

**Review:**

The paper proposes a modular architecture for RL agents aiming at Atari ball games. The hypothesis that authors try to validate is that since there is a lot in common between ball games, such an architecture would demonstrate better learning/transfer to new games after modules are pre-trained on the set o "training" games.
While it is hard to disagree that in general the end-to-end learning approach currently dominating is limited and humans do not learn to solve any of tasks completely end-to-end, I find the presented results, to some extent, self-obvious.
The "cognitive architecture" is so specialized and granular in the way modules are defined that the fact that this approach successfully works is not surprising and it is unclear what conclusions can be drawn based on this result.
I also found it unclear how exactly do authors train the proposed modules and the agent as a whole since in the abstract they write "cognitive architecture capable of end-to-end learning", but later it looks like each of the modules has some loss function associated with it. If so, why couldn't we achieve the same performance using auxiliary tasks (see Jaderberg et al, 2016) and less specialized architecture? Or, alternatively, could we instead rely on model-based RL approaches which, at least in the simple case of Atari ball games, would effectively result into learning very similar functional parts to those that authors propose.

---

### Decision · Program_Chairs · 2018-03-20
**ICLR 2018 Workshop Acceptance Decision**

**Decision:**

Reject

**Comment:**

Based on the reviews, this paper has not been accepted for presentation at the ICLR workshop. However, the conversation and updates can continue to appear here on OpenReview.